# The role of radiation in the Northern Hemisphere troposphere-to-stratosphere transport

Tuule Müürsepp<sup>1</sup>, Michael Sprenger<sup>1</sup>, Heini Wernli<sup>1</sup>, and Hanna Joos<sup>1</sup> Institute for Atmospheric and Climate Science, ETH Zurich, Zurich, Switzerland **Correspondence:** Tuule Müürsepp (tuule.mueuersepp@env.ethz.ch)

**Abstract.** The upper troposphere and lower stratosphere (UTLS) region of the atmosphere is greatly impacted by the exchange of mass and constituents between the two layers. The two-way transport, both stratosphere-to-troposphere (STT) and troposphere-to-stratosphere (TST), has been studied climatologically to quantify the exchange globally or in the context of specific weather systems. However, the local physical processes responsible for the potential vorticity (PV) modification required for crossing the dynamical tropopause have been studied only in detailed case studies.

In this study, we introduce a method of quantifying the role of radiative processes leading to TST and apply this method to 10 years of TST trajectories identified from ERA5 reanalysis. This approach combines a Lagrangian TST identification with a process-specific PV framework. The combination allows us to attribute processes that contribute to the TST. We focus on the period with a significant PV increase of 1 pvu prior to TST and study the contribution of radiation to that increase. Radiation is present in 84% of TST cases, and for every fourth trajectory crossing the 2 pvu dynamical tropopause, radiation is responsible for at least a quarter of the considered 1 pvu increase prior to TST. Focusing on radiatively dominated TST cases, i.e., TST events where radiation contributes with more than 50% to the 1 pvu increase, we find large variability in terms of how PV accumulates, ranging from short to long accumulation times and strongly varying values of the associated PV rates. We show that the high PV rates along the radiatively dominated TST trajectories are mainly produced by temperature tendencies resulting from cloud-top cooling, with maximum values occurring at the top of mixed-phase clouds with high hydrometeor contents. However, a similar magnitude of radiative PV rates can also be produced by vertical gradients in specific humidity. The accumulation time for the 1 pvu increase is determined not only by radiation but mainly by the interplay with other diabatic processes such as turbulence and cloud microphysical processes. In summary, this study provides new insight into the complex interplay of processes and pathways that air parcels experience on their way to the stratosphere.

# 20 1 Introduction

Stratosphere-troposphere exchange (STE) plays an important role in the composition and dynamics of the upper troposphere and lower stratosphere (UTLS). The net mass and constituent flux during stratosphere-to-troposphere (STT, downward) and troposphere-to-stratosphere (TST, upward) transport is determined by the large-scale, wave-driven Brewer-Dobson circulation (Holton et al., 1995). The amplitude of the net transport is small compared to the upward and downward transports, and in the climatological mean it is upward in the tropics and downward in the extratropics (Stohl et al., 2003). However, there are local

30

processes that control exactly when and where the exchange of air across the tropopause occurs (Ploeger and Kunkel, 2024). This study focuses on TST in the extratropics, which is relevant in terms of potential transport of water vapour and pollutants into the lower stratosphere. TST can therefore lead to large positive humidity anomalies in the lower stratosphere (Krebsbach et al., 2006), with important implications for the global radiative balance (Riese et al., 2012).

Both STT and TST exhibit pronounced seasonality and geographical variability (e.g., Škerlak et al., 2014), which is partly shaped by the occurrence of specific weather systems often associated with STE. Specific weather systems that embed the local processes leading to STE have been investigated in case studies and climatologically. For example, these studies looked at STE in relation to monsoon circulations (Gettelman et al., 2004), deep convective systems in the tropics (Corti et al., 2006), overshooting convection in the mid-latitudes (Hegglin et al., 2004), Rossby wave breaking (Sprenger et al., 2007), tropopause folds (Škerlak et al., 2015), and extratropical cyclones (Reutter et al., 2015). Note that these flow systems are not mutually exclusive, since tropopause folds, for example, often co-occur with Rossby wave breaking and extratropical cyclone intensification. STE turns out to be particularly prominent in tropopause folds (Sprenger et al., 2003), and several case studies have quantified STE in deep tropopause folds (e.g., Staley, 1960; Lamarque and Hess, 1994; Bourqui, 2004). While these studies provide valuable information about the meteorological context of STE in specific case studies, they do not climatologically identify the relative role of the physical processes associated with STE.

The dynamical tropopause definition – identifying the tropopause as an isosurface of potential vorticity (PV) – offers an ideal conceptual framework to discuss physical processes related to STE. In the absence of diabatic and frictional processes, PV is a materially conserved quantity (Hoskins et al., 1985), meaning an air parcel cannot cross the dynamical tropopause. In other words, STT and TST require processes that do not conserve PV to occur near the tropopause, which can be explored using the equation for the material PV tendency (see Sect. 2.3 and Eq. 1 for more details). In the UTLS regions, the diabatic and frictional processes that can modify PV include turbulence, radiation and cloud processes involving phase changes of water. Thus far, the role of these processes for STT and/or TST has only been studied on a case-by-case basis (e.g., Zierl and Wirth, 1997; Traub and Lelieveld, 2003; Gray, 2006). For example, Lane and Sharman (2006) showed how convective systems can lead to turbulent mixing in the lower stratosphere due to the breaking of gravity waves induced by deep convection, essentially leading to STT. Lamarque and Hess (1994) emphasized the importance of the latent heating gradient at the tropopause level above the warm sector of a surface cyclone. While they found that this region is responsible for the majority of the cross-tropopause exchange associated with an extratropical cyclone, they also identified TST-conducive regions where large PV tendencies resulted from the radiative heating gradient alone. They argued that, at the tropopause level, even a weak radiative temperature gradient can lead to a strong PV tendency as this also depends on vorticity, which can have local maxima near the tropopause. Gray (2006) conducted a process-based analysis of STT and TST associated with a PV filament and in its environment. They concluded that TST near the filament and south of the ridge upstream was mainly due to radiative processes. Based on previous studies and their own work, they showed that the direction of radiation-induced STE depends greatly on the height to which convection can reach, i.e., the cloud top. Zierl and Wirth (1997) looked at the effect of radiative processes on STE in upper troposphere anticyclones. They attributed TST mass exchange to the vertical gradient of radiative heating and noted that some of it is controlled by the temperature anomaly profile within the anticyclone. Their model did not include clouds or aerosols. A

90

common theme in these studies is the noticeable case-to-case variability of the involved processes, which emphasizes the need for a climatological investigation (Ploeger and Kunkel, 2024).

In this study, we focus specifically on the importance of radiation for TST. A challenging aspect of radiative PV modification is that it can occur even without the presence of a specific weather system (e.g., extratropical cyclones or tropopause folds). Hoskins et al. (1985) estimated that under (relatively) clear-sky conditions in a blocking anticyclone, radiation is responsible for a 1 pvu increase in five days. As Zierl and Wirth (1997) mentioned, in the absence of clouds, the necessary radiative temperature and PV tendencies can arise from the vertical humidity and temperature profiles in the UTLS alone. In such a favourable environment, air parcel trajectories can accumulate PV over longer periods and over greater distances. In the presence of clouds, in particular cold-top cirrus clouds in the extratropical UTLS, the vertical temperature tendency signal is also largely modified by cloud-top cooling and cloud-base heating (Liou, 1986). According to PV theory, this results in an increase in PV above cooling, a decrease in PV between cooling and heating, and an increase in PV below the heating (Hoskins et al., 1985; Wernli and Gray, 2024). In turn, the cloud-radiative response is modified by the temperature, ice water content (IWC), and ice crystal number concentration of the cirrus cloud (Krämer et al., 2016). Similar magnitude PV tendencies are also present in larger mid-latitude cloud systems with cirrus tops within extratropical cyclones. For instance, Chagnon and Gray (2015) demonstrated that the combination of radiation and latent heating processes related to a warm conveyor belt (WCB) cloud created a PV dipole that intensified the PV gradient at the tropopause. Spreitzer et al. (2019) showed in a case study of an extratropical cyclone and the upper-level ridge downstream that radiative PV tendencies are largest close to sharp moisture gradients and exhibit local maxima above clouds. Joos and Forbes (2016) investigated the effect of microphysics on the WCB and the PV modifications due to different processes. Following the trajectories along and above the WCB, they concluded that those within a WCB experience a larger negative PV modification due to cloud processes. However, trajectories above the WCB (and thus above the cloud) are affected by cloud-top cooling and experience a PV gain due to the vertical temperature gradient. Keshtgar et al. (2023) studied the effect of radiation on the evolution of the extratropical cyclone in an idealized setup and concluded that radiation can modify the large-scale flow through the changes in microphysical heating. All these examples of pure cirrus and mixed-phase clouds result in significant variability and uncertainty in the final radiation-induced PV modification. The combination of the "week or so" timescale given by Hoskins et al. (1985) and the relatively weak estimated values of instantaneous PV tendencies due to radiation, compared to turbulence and cloud processes, makes the systematic study of radiation-induced STE challenging. Due to these smaller values, the relevance of radiation may have been overlooked in some case studies. The extended timescale compared to other physical processes requires a different diagnostic approach which would also shed light on the duration of the accumulation period of radiative PV tendencies.

The overall goal of this study is to climatologically quantify the relevance of radiation in TST in the Northern Hemisphere extratropics using a Lagrangian approach. For reasons of computational feasibility, we decided to focus on a recent 10-year period in one hemisphere, assuming that the physics of TST is the same in the Southern Hemisphere and in slightly earlier periods. We opted for a Lagrangian approach because (as we motivated and hypothesised above) for radiative processes to be relevant in TST, modest instantaneous PV rates must accumulate along the flow to reach the substantial PV increase required for TST. Three important methodological decisions are required to address the overall goal:

- 1. How to identify TST events? Here we will use a modified version of the approach established in previous studies (Wernli and Bourqui, 2002; Sprenger and Wernli, 2003; Škerlak et al., 2014).
- 2. How to quantify the role of radiation (and other physical processes) along TST trajectories? Here we also build on an established method, which accumulates PV tendencies from different physical processes along the flow (Joos and Wernli, 2012; Spreitzer et al., 2019; Attinger et al., 2021).
- 3. How to identify the relevant accumulation time period and associated trajectory segment, for TST? Our pragmatic approach is to consider the PV increase along the trajectory from 1 pvu to 2 pvu (since we use the 2 pvu isosurface as the dynamical tropopause). This approach allows us to quantify the processes that increase the PV from typical tropospheric values (<1 pvu) to stratospheric values (>2 pvu). Importantly, this procedure is flow dependent: if radiative PV rates are strong, then the increase of at least 1 pvu happens in a short period and along a short segment. Vice versa for TST events where weaker radiative PV rates accumulate slowly but steadily over longer periods and segments.

The specific research questions addressed in this study are as follows:

- Q1: What is the radiative contribution to TST and what fraction of TST events are dominated by radiation?
- Q2: Where does radiatively dominated TST happen?
- Q3: How and on which time scale does the PV accumulate along the TST trajectories?
  - Q4: What determines the amplitude of the radiative PV rate along the TST trajectory?

To address these questions, we use the aforementioned methodology, which is introduced in more detail in Sect. 2. In Sect. 3 and Sect. 4, we present a 10-year climatology of the identified TST in the Northern Hemisphere, addressing Q1 and Q2. The core of this paper then analyses the exact PV evolution along the TST trajectories (Q3, Sect. 5-6) and studies the vertical profiles of the TST environment in detail to better understand what determines the amplitude of radiative PV rates (Q4, Sect. 7).

#### 2 Data and methods

Our study is entirely based on ERA5 reanalysis (Sect. 2.1). The methods include the identification of TST trajectories (Sect. 2.2), the quantification of diabatic PV modifications along these trajectories (Sect. 2.3), and the approach to quantify the specific role of radiation (Sect. 2.4).

# 2.1 ERA5 reanalyses

We use data from ERA5 (Hersbach et al., 2023), the latest reanalysis dataset from the European Centre of Medium-Range Weather Forecasts (ECMWF). The original data have a 31 km spatial resolution on 137 model levels and an hourly time resolution. This study is limited to the Northern Hemisphere and a 10-year period from 2012 to 2021. Our dataset is interpolated

to a  $0.5^{\circ} \times 0.5^{\circ}$  longitude-latitude grid. It contains standard fields such as temperature, specific humidity, cloud and hydrometeor concentrations for cloud water, cloud ice, rain and snow. It also contains three-dimensional winds, which are used to calculate air parcel trajectories with LAGRANTO 2.0 (Wernli and Davies, 1997; Sprenger and Wernli, 2015). Additionally, we use three-dimensional diabatic heating or cooling rates from ERA5, including total diabatic rates  $Q_{\text{tot}}$ , which corresponds to the sum of the temperature tendencies from all parametrisations, and specifically longwave and shortwave radiative heating or cooling rates  $Q_{\text{lw}}$  and  $Q_{\text{sw}}$ , respectively. These fields are essential for estimating the PV rates due to radiation (PVR<sub>rad</sub>) and other processes along TST trajectories, as explained further in Sect. 2.3.

#### 2.2 TST identification

The TST identification is based on a well-established Lagrangian trajectory approach (Wernli and Bourqui, 2002; Sprenger and Wernli, 2003; Škerlak et al., 2014), but with some modifications. The basic idea of this method is to launch trajectories from a dense grid near the tropopause, trace PV along these trajectories, and then select trajectories as TST candidates that experience an increase in PV from less than 1 pvu to more than 2 pvu within a five-day period. The selected trajectories can then be extended forward and backward in time to: (i) quantify their residence time on either side of the tropopause; (ii) apply a residence time threshold criterion to exclude potentially spurious TST events with a very short residence time in the stratosphere; (iii) investigate the origin of the TST trajectories; and (iv) assess the physical processes that occurred along these trajectories before crossing the tropopause.

This study only considers TST in the extratropical Northern Hemisphere. The trajectories therefore start from an equidistant grid of  $50 \,\mathrm{km} \times 50 \,\mathrm{km}$  within an area spanning from  $20^\circ\mathrm{N}$  to  $90^\circ\mathrm{N}$ . In the vertical, the trajectories start at  $20 \,\mathrm{hPa}$  intervals from  $610 \,\mathrm{hPa}$  to  $210 \,\mathrm{hPa}$ , i.e.,  $21 \,\mathrm{levels}$  in total. The 1-day forward trajectories are initiated every  $24 \,\mathrm{hours}$  from  $00 \,\mathrm{UTC}$  on 1 January  $2012 \,\mathrm{to}$  00 UTC on 31 December 2021. This 10-year period is shorter than that investigated in the previous TST climatology by Škerlak et al. (2014). This is because of the time-consuming additional process analysis performed along the TST trajectories (see Sect.  $2.3 \,\mathrm{and}$  2.4). The main aim of this study is not to provide a long-term climatology of TST, but rather to obtain a sufficiently long dataset to robustly quantify the role of radiation in TST.

We set the stratospheric residence time threshold to only 2 h in this study. This means that we will also study TST trajectories that only make a short excursion into the stratosphere. Such events can still be significant for atmospheric composition if the TST is associated with turbulence, which leads to rapid mixing of the original tropospheric air with the stratospheric environment. Furthermore, we applied a second filtering criterion to prevent the occasionally erroneous interpretation of PV values greater than 2 pvu as stratospheric. For instance, latent heating in clouds and radiative heating near strong boundary layer inversions can lead to PV production (e.g., Wernli and Gray, 2024) exceeding the 2 pvu threshold used here to define the tropopause. We use the three-dimensional labelling of PV structures, introduced by Sprenger et al. (2003) and Škerlak et al. (2015) for this filtering. This labelling distinguishes stratospheric PV from isolated diabatically and frictionally produced PV anomalies in the troposphere that exceed 2 pvu. In our case, this filter was particularly important for removing trajectories entering the very stable boundary layer with high PV over Greenland. The TST trajectories remaining after this filtering were then extended 96 h backwards in time. This results in 5-day TST trajectories with the crossing of the tropopause taking place

any time during the final 24 h. The PV modification by radiation and other diabatic processes is then quantified along these 160 TST trajectories.

Before explaining how the TST contributions of several processes are quantified, we introduce the concept of a "TST duration". When quantifying the role of different processes for the PV increase along TST trajectories, we must specify the segment of the trajectory along which this quantification is performed. One option would be to use a fixed time period or spatial distance (e.g., the 6 h prior to TST, or the  $1000 \, \mathrm{km}$  before crossing the tropopause), but these approaches would not be flexible enough for achieving the goals of this study. Therefore, we determine the last time that the air parcel had a PV of 1 pvu before crossing the tropopause for each TST trajectory,  $t_{1\mathrm{pvu}}$ , in addition to the time of crossing the tropopause (i.e., the TST time step,  $t_{2\mathrm{pvu}}$ , see also Fig. 1). The period between these two time steps is defined as the TST duration,  $\mathcal{T}_{\mathrm{TST}}$ , which, as shown below, can vary from a few hours to four days. Therefore, this approach works well in cases with rapid PV production prior to TST and in other cases where the PV increase occurs over longer periods and trajectory segments.

### 170 2.3 Process-specific diabatic PV rates

The total PV rate (PVR), i.e., the material derivative of PV, is given by

$$PVR = \frac{D}{Dt}PV = \frac{1}{\rho} (\boldsymbol{\eta} \cdot \nabla Q + \nabla \times \mathbf{F} \cdot \nabla \theta), \qquad (1)$$

where  $\rho$  denotes air density,  $\eta$  the absolute vorticity vector,  $Q = D\theta/Dt$  the diabatic heating rate, i.e., the material derivative of potential temperature, and  ${\bf F}$  the non-conservative forces (Hoskins et al., 1985; Wernli and Gray, 2024). Thus, material changes in PV can occur due to two types of physical processes. The first term in Eq. 1 represents the processes that lead to potential temperature changes due to diabatic processes ( $Q_{\rm tot}$ ), including latent heating due to cloud-microphysical processes and convection, turbulent mixing, and radiative processes ( $Q_{\rm rad} = Q_{\rm lw} + Q_{\rm sw}$ ). The second term is related to non-conservative forces that lead to PV modification due to momentum tendencies associated with turbulence, friction and convection. Applying Eq. 1 for each individual process  $Q_i$  where i denotes the specific process, yields three-dimensional fields of process-specific PV rates, PVR $_i$ .

This methodology was introduced by Joos and Wernli (2012) and later used in studies where the individual process-based heating rates and non-conservative forces were output from dedicated numerical model simulations (e.g., Spreitzer et al., 2019; Attinger et al., 2021). Here, for the first time, we apply this approach to ERA5 data, where only a selection of process-based fields is available. As mentioned in Sect. 2.1, we use the ERA5 variables mean temperature tendency due to short-wave ( $Q_{sw}$ ) and long-wave radiation ( $Q_{lw}$ ) to quantify the radiative contributions to the diabatic heating or cooling and the associated PV rates. Furthermore, the mean temperature tendency due to all parametrisations ( $Q_{tot}$ ) is also provided in ERA5. From these individual heating rates, process-specific PV rates can be calculated for radiation and its longwave and shortwave components (PVR<sub>rad</sub>=PVR<sub>lw</sub>+PVR<sub>sw</sub>), for the sum of all diabatic heating contributions based on  $Q_{tot}$  (PVR<sub>diab</sub>), and consequently for all non-radiative diabatic heating tendencies, which are due to cloud microphysics, convection, and turbulence, as the difference of the two former PV rates (PVR<sub>diab</sub>-PVR<sub>diab</sub>-PVR<sub>rad</sub>). In addition, we could, in principle, calculate the corresponding

PV rate from the momentum tendencies ( $PVR_{mom}$ ). Hence, the total PV rate ( $PVR_{tot}$ ) can be written as:

$$PVR_{tot} = PVR_{diab} + PVR_{mom} + \underbrace{\mathcal{R}}_{Residual} = (PVR_{rad} + PVR_{diab-rad}) + PVR_{mom} + \underbrace{\mathcal{R}}_{Residual}. \tag{2}$$

We calculate  $PVR_{tot}$  approximately as the change in PV along the trajectory during a given time step (1 h in ERA5). Due to numerical issues in the computation of the trajectories and the various PV rates, we expect a (small) residual to close the budget, in agreement with earlier studies (e.g., Spreitzer et al., 2019). Another factor that most likely contributes to the residual is that the trajectories run over multiple 12-hour data assimilation windows (ECMWF Copernicus Climate Change Service, 2025) that lead to potential discontinuities in the temperature tendencies and associated PV rates twice a day. For completeness, it has to be noted that in this study we do not quantify explicitly the contribution of the momentum tendencies, which means that both the  $\mathcal{R}$  and  $PVR_{mom}$  are unknowns, whose sum corresponds to the difference of  $PVR_{tot}$  and  $PVR_{diab}$ . The fields of  $PVR_{sw}$ ,  $PVR_{lw}$ , and  $PVR_{diab}$  can be traced along trajectories of interest, which enables a Lagrangian quantification of  $PVR_{mod}$  processes for TST, as explained in the next subsection.

# 2.4 Process attribution along TST trajectories

As the last step of the methodology, we accumulate the radiative PV rates along the TST trajectories during the TST duration,  $\mathcal{T}_{TST}$ . As introduced above,  $\mathcal{T}_{TST}$  corresponds to the period of PV increase from less than 1 pvu to more than 2 pvu for every individual TST trajectory, as shown in the example in Fig. 1. The corresponding increase of at least 1 pvu prior to crossing the tropopause is regarded as a 'significant PV gain' and  $\mathcal{T}_{TST}$  therefore a meaningful period to quantify the role of radiation for TST. The 1 pvu increase is calculated by tracing the PV values backwards from the TST time step and noting the first time step where PV  $\leq$  1 pvu ( $t_{1pvu}$ ).

The process-specific accumulation of PV rates along the trajectories follows the previous studies about Lagrangian PV changes (Joos and Wernli, 2012; Spreitzer et al., 2019; Attinger et al., 2021). We use the notation of these earlier studies and introduce APV to denote accumulated PV rates along trajectories. More specifically, the accumulated PV rate due to radiation,  $APV_{rad}$ , during the period  $\mathcal{T}_{TST}$  is calculated as

$$APV_{rad} = \sum_{k=0}^{n-1} PVR_{sw}(\mathbf{x}(t_k), t_k) \Delta t + \sum_{k=0}^{n-1} PVR_{lw}(\mathbf{x}(t_k), t_k) \Delta t,$$
(3)

where k=0 and k=n correspond to the temporal indices of the hourly trajectory positions just before  $t_{1\mathrm{pvu}}$  and just after  $t_{2\mathrm{pvu}}$ , and  $\mathbf{x}(t)$  denotes the trajectory. Coloured lines in Fig. 1 show the time evolution of  $\mathrm{PVR_{rad}}$  and  $\mathrm{APV_{rad}}$  for an example trajectory, together with the overall PV evolution. The figure illustrates that with this approach,  $\mathrm{APV_{rad}}$  integrates the role of radiation for the PV evolution during the period  $\mathcal{T}_{\mathrm{TST}}$  and therefore the PV evolution (and potential contributions from radiative processes) prior to  $t_{1\mathrm{pvu}}$  are not accounted for. The same procedure is also applied to the total diabatic PV rate,  $\mathrm{PVR_{diab}}$ , which yields  $\mathrm{APV_{diab}}$ . To then calculate the contribution of radiation to the 1 pvu increase, we consider the following PV budget during  $\mathcal{T}_{\mathrm{TST}}$ :

$$APV_{rad} + APV_{diab-rad} + APV_{mom} + \mathcal{R} = APV_{tot} \ge 1 \text{ pvu}. \tag{4}$$

Figure 1. Schematic to explain the elements of the methodology. Time axis shows forward and backward calculation of the trajectory from time  $0\,\mathrm{h}$ . The black line shows the PV evolution along a TST trajectory, the vertical black dashed lines indicate  $t_{1\mathrm{pvu}}$  and the TST time step  $(t_{2\mathrm{pvu}})$ . The period in between corresponds to the TST duration,  $\mathcal{T}_{\mathrm{TST}}$ . The red line shows the radiative PV rate  $(\mathrm{PVR_{rad}})$  with the dashed red line noting the time of maximum radiative PV production during  $\mathcal{T}_{\mathrm{TST}}$ . The yellow line shows the accumulated PV change due to radiation during  $\mathcal{T}_{\mathrm{TST}}$ . An example of the residence time filtering can be seen at  $t=8\,\mathrm{h}$ , when PV crosses the  $2\,\mathrm{pvu}$  threshold for only one hour and it is therefore not considered as the TST time step. Note the different axes for PV and PVR on the left and right of the diagram, respectively.

Here,  $APV_{\rm tot}$  is diagnosed from the PV values along the trajectories (PV difference between times  $t_{\rm 1pvu}$  and  $t_{\rm 2pvu}$ ), as described above, and the sum of the momentum tendency contribution and the residual can be calculated as the difference between  $APV_{\rm tot}$  and  $APV_{\rm diab}$ . Finally, we can express the definition of the radiative contribution to TST for every trajectory by the ratio

$$f_{\rm rad} = \frac{\rm APV_{\rm rad}}{\rm APV_{\rm tot}},$$
 (5)

which can attain negative values in case radiation works against the PV production during  $\mathcal{T}_{TST}$ . In cases of strong radiative PV production, the ratio can reach values up to 100% or even larger if other processes work against radiation. In the next section, we introduce the notion of "radiatively dominated TST", which refers to TST trajectories along which radiative PV tendencies produce the majority ( $f_{rad} > 50\%$ ) of the required accumulated PV increase of 1 pvu.

Figure 2. (a) Statistical distribution of  $APV_{rad}$  for all TST trajectories (bin width = 0.01) and a 4-category classification, and (b) the seasonal cycle of the four TST categories shown in (a) and defined in Table 1. The bars in (b) show each category's contribution to the given month (in %, y-axis on the left), and the dashed lines show absolute numbers of TST trajectories (in  $10^6$ , y-axis on the right).

### 3 Radiative PV contribution to TST




The final TST dataset for the 10-year period includes 35.4 million trajectories following the initialisation criteria and the filtering (Sect. 2.2). Firstly, we conduct a statistical analysis of the radiative contribution to TST ( $f_{\rm rad}$ ), followed by a discussion of the geographical distribution of TST and, in particular, of radiatively dominated TST.

Figure 2a shows the statistical distribution of  $APV_{\rm rad}$  values for all TST trajectories. Note that because  $APV_{\rm tot} \sim 1~\rm pvu$ , a value of  $APV_{\rm rad} = 0.5~\rm pvu$  corresponds roughly to a value of  $f_{\rm rad} = 50\%$ . A certain fraction of TST trajectories (16%) has a negative PV contribution from radiation, but the majority (59%) of TST trajectories show a small positive contribution from radiation with  $APV_{\rm rad}$  values between 0 and 0.25 pvu. The distribution is strongly skewed with a long right tail. 17% of the TST trajectories have a moderate contribution from  $APV_{\rm rad}$  (between 0.25 and 0.5 pvu), and, finally, for 8% of the TST trajectories, radiation is the dominant process leading to TST, with  $APV_{\rm rad} > 0.5~\rm pvu$ . In summary, Fig. 2a shows that radiation is contributing positively to most of the TST trajectories, although the contribution itself tends to be small, as the majority of them have  $0~\rm pvu < APV_{\rm rad} < 0.25~\rm pvu$  (yellow). In the following, we will refer to these four parts of the distribution as categories I to IV (see also Table 1), where category IV is the most interesting one for this study, and referred to in the following as "radiatively dominated TST".

Figure 2b presents the seasonality of TST of the four categories. In total, TST is more frequent in winter, which is consistent with previous studies (e.g., Škerlak et al., 2014). Categories with a positive radiative PV contribution exhibit a similar season-

<sup>&</sup>lt;sup>1</sup>The distribution of radiative PV changes along trajectories is skewed because of the selection bias related to our choice of focusing on TST trajectories. If a similar analysis was performed along randomly selected trajectories in the upper troposphere, the radiative PV changes are expected to be symmetrically centred at zero.




Table 1. The definition of the four TST categories based on the radiative contribution  $APV_{\rm rad}$ , as also seen in Fig. 2a.

| Category I   | $APV_{rad} \le 0 pvu$                                       | radiation works against TST                               |
|--------------|-------------------------------------------------------------|-----------------------------------------------------------|
| Category II  | $0\mathrm{pvu} < \mathrm{APV_{rad}} \le 0.25\mathrm{pvu}$   | radiation has a small positive contribution to TST        |
| Category III | $0.25\mathrm{pvu} < \mathrm{APV_{rad}} \le 0.5\mathrm{pvu}$ | radiation has a considerable positive contribution to TST |
| Category IV  | $\mathrm{APV_{rad}} > 0.5\mathrm{pvu}$                      | radiation is the dominant process leading to TST          |

ality with a maximum in winter and a minimum in summer. Category I with a negative radiative contribution has a reversed seasonal pattern. In winter, this category accounts for about 10%, and in summer, its share increases to 30%. This suggests that in summer radiation tends to work more often against PV production along the TST trajectories. Separating the radiative contribution into  $APV_{lw}$  and  $APV_{sw}$  (not shown) reveals that half of the negative contribution still comes from the longwave radiation. This indicates that shortwave radiation has a role in PV modification in summer, but its effect alone is insufficient to explain the seasonal cycle of category I.

# 4 Spatial climatology of (radiatively dominated) TST

Figure 3 gives the overview of the spatial climatology of the identified TST. The top row shows that total TST is most frequent in the storm track regions and over Scandinavia, with an additional local maximum over central Asia in JJA. The North Atlantic storm track signal extends from central North America to the Ural Mountains, and the North Pacific storm track signal from the Japan Sea to the Gulf of Alaska. These distributions are in agreement with previous Lagrangian studies of TST that have also found TST maxima in the storm track regions (Stohl, 2001; Wernli and Bourqui, 2002; Sprenger and Wernli, 2003; Škerlak et al., 2014).

Throughout the seasons, the most prominent local maximum is located at the south tip of Greenland. This peak is also present in most of the aforementioned studies, but its amplitude depends on the chosen residence time threshold, which was 24 h or more in the previous studies. With the much shorter residence time threshold here, the local maximum becomes stronger, indicating that the residence time of most TST events over the south tip of Greenland is short. The same holds for the central Asian summer peak, which was not present in previous TST climatologies. We speculate that this peak over the Pamir and Tien Shan Mountains is a result of a combination of the northward shift of the subtropical jet in summer (Schiemann et al., 2009) and the fact that the Tien Shan Mountains experience a monsoon-like climate (Jin et al., 2024). The crossing of the tropopause in that specific region is likely possible because the subtropical jet can provide the necessary turbulent environment, or the monsoon clouds and associated humidity gradients can contribute to strong diabatic temperature tendencies. The fact that no such signal is discernible further south over the Himalayas in summer (when it could be expected due to the monsoon) or in winter (when the subtropical jet is south of the Tibetan Plateau) is most likely because trajectories were initialised only poleward of 20°N. Martius (2014) calculated seven-day backward trajectories from the subtropical jet in the Northern Hemisphere and their results suggest that in winter a significant amount of the trajectories ending in the subtropical jet over Asia start south of  $20^{\circ}$ N.

Figure 3. Seasonal climatologies of the density of TST trajectories at the time of crossing the tropopause from 2012-2022 in the Northern Hemisphere (from left to right, DJF, MAM, JJA, and SON). The density is calculated as trajectories  $\rm km^{-2}$  for the 10 years. The first row shows all TST categories together and the second row the radiatively dominated category IV. Note the factor 7 difference between the colour bars. The third row shows the fraction of the second to the first row, i.e., the percentage of TST in category IV (values are not shown where the total TST density is below 0.01 trajectories  $\rm km^{-2}$ ). The black contour in the third row shows the 8% line representing the climatological average fraction of radiatively dominated cases as seen in Fig. 2a.

The second row in Fig. 3 shows TST in category IV, and the third row its fraction of total TST. Since overall 8% of the TST cases belong to this category, in regions where the relative fraction is higher than 8% (note the black contour in the third row), the radiatively dominated category IV is more frequent than on average. In comparison to total TST, the trajectory densities of category IV in winter show even more distinct maxima over the North Atlantic and North Pacific basins with the North Atlantic signal shifted over Greenland. In relative terms, the higher than average category IV densities are located slightly northward



Figure 4. Overview of the properties of the TST categories: (a) probability density function of average  $PVR_{rad}$  (solid lines) and  $PVR_{rad,MAX}$  (dashed lines) for categories II-IV; (b) histograms of TST duration ( $\mathcal{T}_{TST}$ ), for all categories in 1-hourly bins. The solid lines in (b) show the fraction [%] of trajectories in that category to all trajectories in the corresponding bin. (a) is clipped at 0.03 pvu to show the differences among the categories.

of the maxima in the full TST climatology; largest values (of 15% and above) are found poleward of  $60^{\circ}$ N. In spring, the category IV distribution is more uniform with lower densities than in winter (consistent with Fig. 2b) and relative contributions of 5-10% everywhere. The summer map reveals that many central Asia cases seen in the total climatology belong to category IV since the relative contribution there exceeds 25%. As to why the mountain ranges enable more radiative TST in that region would need a more in-depth investigation, which is beyond the scope of this study. Everywhere else, the category IV fraction is lower than the climatological average in summer. The autumn distribution appears to be a combination of the summer and winter regimes, with also a clear signal in central Asia and large relative contributions to TST at high latitudes.

The main insights from the spatial analysis are that (i) this TST climatology agrees fairly well with earlier longer-period climatologies (based on older reanalyses), (ii) TST in category IV tends to occur preferentially at higher latitudes in all seasons except summer, and (iii) except for local maxima over Greenland and in central Asia discussed above, radiatively dominated TST occurs rather uniformly in the entire extratropical Northern Hemisphere.

### 5 What determines the TST duration?

We continue to analyse the accumulation of  $PVR_{rad}$  in the TST duration time window ( $\mathcal{T}_{TST}$ ) during which the 1 pvu increase happens. First, we take a look into how the average  $PVR_{rad}$  (hourly radiative PV rates averaged along each trajectory) and  $PVR_{rad,MAX}$  (maximum hourly radiative PV rate along each trajectory) during  $\mathcal{T}_{TST}$  differ among the categories. In addition, we show the statistical distributions of  $\mathcal{T}_{TST}$  for the four categories. Figure 4a shows that for categories II-IV with a positive radiative contribution, most of the average and maximum values of  $PVR_{rad}$  are below  $0.03 \, pvu/h$ . The most frequent average value (solid lines) is near  $0.005 \, pvu/h$  for the smallest contribution category II (yellow line) and increases up to  $0.009 \, pvu/h$ 





for the radiatively dominated category IV (red line). We see the same behaviour for  $PVR_{\rm rad,MAX}$  at slightly higher values. The distribution also gets flatter towards category IV. All in all, and not surprisingly, the more radiation is contributing to TST, the higher the average values of  $PVR_{\rm rad}$  and  $PVR_{\rm rad,MAX}$ . However, the values within each category vary strongly between almost 0 to  $0.2\,\mathrm{pvu/h}$  (not shown here, see distributions in Fig. 5a). Considering the average  $PVR_{\rm rad}$ , this indicates that the PV is produced over very different time durations. Also, we conclude that  $PVR_{\rm rad,MAX}$  is unlikely to be the determining factor of the radiative contribution, given its comparatively low values.

To confirm this, Fig. 4b shows the distribution of TST duration ( $\mathcal{T}_{TST}$ ) among the categories. We see that categories I and II tend to have shorter  $\mathcal{T}_{TST}$  with pronounced peaks below  $\mathcal{T}_{TST} = 10\,\mathrm{h}$ , whereas categories III and IV show a flatter distribution with almost equal frequencies from 10 to 100 h. (The peaks in the distributions that appear every 12 h correspond to the ERA5 assimilation windows, as mentioned in Sect. 2.4.) In another perspective, for  $\mathcal{T}_{TST} = 70\,\mathrm{h}$  and above, the fraction of category III-IV trajectories is larger than for categories I-II, meaning that if the PV increase for a TST trajectory takes longer, it is more likely to experience the increase due to radiative processes.

We now focus on category IV cases and analyse the possible TST pathways in terms of  $\tau_{TST}$ , average  $PVR_{rad}$ , and  $PVR_{rad,MAX}$ . Figure 5a shows a 2D histogram of  $\tau_{TST}$  and average  $PVR_{rad}$  coloured by  $PVR_{rad,MAX}$ . The hyperbolic shape indicates that  $\tau_{TST}$  and  $PVR_{rad}$  are to a certain degree anti-correlated. But still, a small average  $PVR_{rad}$  can exhibit any TST duration from 20 to 120 h, and a short TST duration can accompany a wide range of average  $PVR_{rad}$  values. The marginal distributions in Fig. 5 reveal that most events have small average  $PVR_{rad}$  values, in agreement with Fig. 4a. The histogram covers average  $PVR_{rad}$  values of up to 0.2 pvu (compared to only 0.03 pvu in Fig. 4a) to show that these high values exist but they are very rare.  $PVR_{rad,MAX}$  tends to be larger along the inner edge of the hyperbolic distribution, but it seems to be a poor discriminating factor for determining  $\tau_{TST}$ . We can conclude from the distribution that radiatively dominated TST can accumulate PV in a large variety of ways.

To illustrate some of the different pathways, Fig. 5b-e show example TST trajectories. The first example (Fig. 5b) has a long TST duration, small average  $PVR_{rad}$  and small  $PVR_{rad,MAX}$  (exact values are given within the panels). The second example (Fig. 5c) has also a long TST duration but slightly larger average  $PVR_{rad}$  and  $PVR_{rad,MAX}$ . Figures 5d,e show short  $\mathcal{T}_{TST}$  cases with medium and large average  $PVR_{rad}$  and  $PVR_{rad,MAX}$ , respectively. Maps with the pathways of these trajectories and vertical profiles along them are given in Fig. 6. It is interesting to take a closer look at the exact PV evolution for each case and to see what kind of vertical profiles produced the PVRs relevant for TST.

Figure 5b shows a steady increase of PV during  $\mathcal{T}_{TST}$  and we see that the  $PVR_{lw}$  signal is small but always positive.  $PVR_{sw}$  reveals the diurnal cycle and tends to contribute to PV destruction whenever present. The steadiness of the  $PVR_{lw}$  signal indicates a rather stationary environment over almost four days. The corresponding Fig. 6a,b show the trajectory starting from the south-west of the Iberian Peninsula, moving first south and then over Europe and the Mediterranean Sea. The vertical profile reveals no clouds at the height of the trajectory or just below it, but the trajectory follows a constant vertical humidity gradient at the height of 300 to 200 hPa. The case in Fig. 5c and Fig. 6c,d exhibits a similar steady increase of PV over 81 h but radiation contributes only during about half of  $\mathcal{T}_{TST}$ . The rates are 10 times larger than for the previous case and indicate that the trajectory passed two regions of high  $PVR_{rad}$ . The trajectory stayed over North America for the five-day period and moved

Figure 5. (a) The central panel shows the 2D histogram of average  $PVR_{rad}$  and  $\mathcal{T}_{TST}$  for the radiation dominated category IV, coloured by the average  $PVR_{rad,MAX}$  value for every bin. On the top and to the left of the histogram are the marginal distributions of the binned data (as in Fig. 4a,b for the average PVR and  $\mathcal{T}_{TST}$ , respectively). (b-e) show example TST trajectories marked with red dots in the central panel. The time series show PV (black),  $PVR_{lw}$  (blue) and  $PVR_{sw}$  (orange). Dashed lines mark  $t_{1pvu}$  and the TST time step ( $t_{2pvu}$ ); the distance between them represents  $\mathcal{T}_{TST}$ . Example trajectories (b), (d) and (e) were started at 00 UTC on 10 December 2020 and (c) at 00 UTC on 12 December 2020, and the time on the x-axis is relative to this starting time. Note the different axis for  $PVR_{rad}$  among the examples.

above two distinct clouds with strong moisture gradients that also produced the stronger PVRs. Figure 5d and Fig. 6e,f show a fast TST ( $\mathcal{T}_{TST} = 24\,\mathrm{h}$ ) with two episodes of strong PV increase: the first one during the first 12 h of  $\mathcal{T}_{TST}$  and the second one immediately before crossing the tropopause. Radiation contributed strongly to the first PV increase but had little influence on the second one. It has to be kept in mind that although we are focusing on the radiatively dominated category IV, other PV non-conservative processes still affect the PV evolution. The  $PVR_{rad,MAX}$  occurred on top of a vertically extended cloud, which shows a strong cloud-top cooling (Fig. 6f). This trajectory also stands out as a two-way exchange case. The trajectory

Figure 6. (a, c, e, g) Maps of the four example TST trajectories presented in Fig. 5 in the respective order and (b, d, f, g) vertical profiles along these trajectories. Trajectories are coloured by their PV values [pvu]. The red circles are for  $t_{1\text{pvu}}$ , crosses for maximum PVR and squares for the TST time step ( $t_{2\text{pvu}}$ ). Vertical profiles show specific humidity [mg kg<sup>-1</sup>] in colour,  $Q_{rad}$  ( $-0.2 \, \text{K} \, \text{h}^{-1}$ ) in green contours and PVR<sub>rad</sub> ( $0.01 \, \text{pvu/h}$ ) in yellow contours. The dotted regions mark clouds with combined IWC and LWC (liquid water content) of at least  $10 \, \text{mg} \, \text{kg}^{-1}$ . The gray shading denotes orography.








started in the stratosphere over Europe, lost some of its PV (above mountains, not due to radiation) and then quickly gained it again over a cloud system over the North Pacific. Had we applied a residence time criterion of, e.g.,  $24 \, h$ , this case would not have been included in this analysis. However, for the purpose of this study, it is an important example that radiation can 'push' the once stratospheric air, which has transiently crossed the tropopause, back to the stratosphere. The final case (Fig. 5e and Fig. 6g,h) exhibits an even shorter  $\mathcal{T}_{TST}$  and the PV evolution reveals again episodes with large PV increases.  $PVR_{lw}$  contributes to the initial PV increase with a rate of  $0.6 \, pvu/h$ . This exceptional rate is related to a very strong gradient of the vertical temperature tendency due to a cloud-top cooling over another vertically extended cloud over North America related to a warm conveyor belt (Fig. 6h).

In summary, these examples show that radiation can contribute to TST in very different ways. This is also the reason why even within a single category with a similar radiative contribution to the PV increase, determining a common environment proves to be difficult. Both the large-scale and the small-scale environments that produce the instantaneous radiative PV rates vary greatly, albeit leading to the same result, i.e. TST. This and the simple fact that  $f_{\rm rad}$  rarely exceeds 80% indicate that other processes modify PV at the same time, and some of them contribute to TST also for radiatively dominated cases. We therefore investigate next the interplay between radiation and other PV-producing processes that provide the PV change required for TST.

### 6 The interplay of radiation and other processes in modifying PV

The TST duration,  $\mathcal{T}_{TST}$ , differs even for very similar  $APV_{rad}$  values and thus it is most likely strongly influenced by other processes. We therefore quantify the PV production by non-radiative temperature tendencies,  $APV_{diab-rad}$ , and by momentum tendencies,  $APV_{mom} + \mathcal{R}$ , compared to the contribution from radiation,  $APV_{rad}$ .

Figure 7 shows how these three types of processes contribute to TST in general, and in category IV in particular. In Fig. 7a, we compare two subsets of category IV with short and long  $\mathcal{T}_{TST}$ . First, we notice that the contributions from radiation (in red) are similar for both subsets, which confirms the results from the analysis before that the radiative contribution itself does not strongly determine  $\mathcal{T}_{TST}$ . The momentum tendency contributions (green) are mainly positive and their distribution is shifted to lower values for longer  $\mathcal{T}_{TST}$ . In contrast, the distribution of the non-radiative temperature tendency contributions (black) shifts from largely negative values (peak at  $-0.1~\mathrm{pvu}$ ) for short  $\mathcal{T}_{TST}$  to an almost symmetric distribution around 0 pvu for long  $\mathcal{T}_{TST}$ . Both these shifts indicate that the longer the  $\mathcal{T}_{TST}$ , the smaller are the absolute contributions from other processes. Or in other words,  $\mathcal{T}_{TST}$  can be long if the relatively fast processes such as turbulence and latent heating in clouds contribute less in absolute terms. Another interpretation of Fig. 7a is that  $\mathcal{T}_{TST}$  increases when PV modification due to non-radiative temperature tendencies deviates strongly from zero, because this contribution is typically negative, i.e., opposing TST for category IV events.

Considering the three processes in a joint distribution histogram for TST events in all categories (Fig. 7b) helps to explain all the combinations that enable long TST durations. Firstly, we notice two gradients of  $\mathcal{T}_{TST}$  (colours) in the plot. One gradient appears across the  $APV_{rad}$  isolines (dash-dotted lines), meaning the larger the radiative contribution, the longer  $\mathcal{T}_{TST}$ . A



Figure 7. (a) Histograms of process contributions to the PV increase during  $\mathcal{T}_{TST}$  for radiatively dominated category IV cases with short  $\mathcal{T}_{TST}$  (12 to 24 h, colour filled, 387'109 cases) and with long  $\mathcal{T}_{TST}$  (84 to 96 h, lines, 322'535 cases); bin width = 0.03 pvu. (b) 2D histogram (joint distribution) of the full TST dataset with  $APV_{diab-rad}$  on x-axis and  $APV_{mom} + \mathcal{R}$  on the y-axis, coloured by the average  $\mathcal{T}_{TST}$  for each bin, with bin width = 0.07 pvu for both variables. The solid white lines represent the distribution density with the numbers indicating the number of TST events per bin. The red dot marks the origin of the plot and white dashed lines mark the quadrants (1st-4th) to guide the eye. As the sum of the three contributions always adds up to 1 pvu, the dash-dotted lines represent constant values of  $APV_{rad}$  (0, 0.25, and 0.5 pvu). Therefore, the lines also separate the data into the four previously introduced categories (Table 1).

second gradient, about perpendicular to the first one, is visible within the radiation dominated category IV (below the lowermost dash-dotted line), showing the importance of the contribution from other processes. To explain this, we take a closer look at the region in the scatter plot around the origin, (0,0), and the quadrants around it. In every quadrant near the origin,  $\mathcal{T}_{TST}$  is long because the other processes contribute very little to the TST. In the first and second quadrants (but further away from the origin) the  $\mathcal{T}_{TST}$  values are shorter compared to the other two quadrants, indicating that whenever momentum tendencies contribute (positive  $APV_{mom} + \mathcal{R}$ ), the shorter  $\mathcal{T}_{TST}$  gets and even shorter if also  $APV_{diab-rad}$  is positive (1st quadrant). The third quadrant (negative  $APV_{mom} + \mathcal{R}$  and  $APV_{diab-rad}$ ) shows particularly large  $\mathcal{T}_{TST}$ , which indicates that radiation needs more time to build up the 1 pvu while also compensating for the other processes working against TST. In the fourth quadrant (negative  $APV_{mom} + \mathcal{R}$  and positive  $APV_{diab-rad}$ ) the duration is uniformly large as the other processes balance each other out and radiation alone has to build up the 1 pvu. Only when  $APV_{diab-rad}$  becomes larger than 1 pvu, does the TST duration become shorter again.

This analysis confirms that PV production due to radiation is a relatively slow process. When the radiatively dominated TST category IV shows faster PV build-up, it is caused by other physical processes contributing to the PV increase. The difference between the parts of category IV in the 2nd and the 4th quadrants indicates that the faster PV gain is mostly due to turbulent processes rather than from other non-radiative temperature tendencies.

This also supports the conclusion from Sect. 5, that, for most cases, a combination of different processes is needed to produce the PV necessary for air parcels entering the stratosphere. Therefore, a systematic analysis of the environment that


TST air parcels are moving through becomes challenging. The properties of this environment can change drastically depending on what processes are involved, and a single trajectory can be influenced by the many different PV-producing and destroying processes along its way. However, as the focus of this work lies on the importance of radiation for PV production, in the following section we will analyse which atmospheric conditions determine the magnitude of the radiative PV rates and in which environments this process can become the dominant one for PV production leading to TST.

# 7 Vertical profiles associated with different intensities of positive $PVR_{rad}$

The category IV case study analysis in Sect. 5 revealed a large variability in the time evolution of  $PVR_{rad}$  along TST trajectories, which raises the question of what determines the instantaneous  $PVR_{rad}$  at a certain location. Since radiation is treated in the model in vertical columns, we perform a column-based analysis of the vertical profiles of temperature, humidity, and clouds, associated with different values of  $PVR_{rad,MAX}$  for TST category IV (2.9 million trajectories). The distribution of  $PVR_{rad,MAX}$  in this category is given by the red dashed line in Fig. 4a.

### 7.1 Categorization of vertical profiles

To understand the origin of very high values of PVR<sub>rad</sub> we need to examine the vertical profile of cloud and humidity content as these high rates are expected to be linked to cloud-top cooling or regions with strong humidity gradients. As a first step, we consider possible combinations of how PVR<sub>rad</sub> can take negative or positive values in a vertical profile. Figure 8a shows an idealized schematic of a layer that absorbs/emits radiation (in gray) in the atmosphere. In the absence of the cloud, the layer would be in radiative balance, i.e., it emits an equal amount of radiation as it absorbs from the layers above and below. The same balance applies for the layers above and below. In general, this absorbing/emitting layer represents a cloud, but it could 405 also be a layer with increased humidity. In the following discussion, we refer to this layer as 'a cloud'. First, we focus on the longwave signal. The upper part of the profile exhibits cloud-top cooling and the lower part cloud-base heating (because the cloud layer 'disturbs' the radiative balance of cloud-free conditions). The corresponding temperature tendency profile leads to three layers with alternating signs of PVR<sub>lw</sub>. PVR<sub>rad</sub> is positive above the maximum cooling and below the maximum heating, as indicated by the red plus signs. In between, the signal is negative (blue minus sign). For the shortwave signal, the heating maximum is within the cloud. The values of  $PVR_{sw}$  are therefore negative above and positive below the cloud. The combination of the radiative bands leads to four distinct layers in terms of PV production, as indicated in Fig. 8a. Layer 1 is above the cloud and can exhibit a positive  $PVR_{rad}$  if  $|PVR_{lw}| > |PVR_{sw}|$ . In layer 2, PV production is not possible because both radiation bands produce a negative PVR. In layer 3,  $PVR_{rad}$  is positive only if  $|PVR_{sw}| > |PVR_{lw}|$ . And finally, layer 4 always exhibits radiative PV production. It is important to note that the position of the heating and cooling maxima depend on the exact profiles of moisture and hydrometeor content of the cloud (and, in reality, also on their shape and size distribution (Liou, 1986; Krämer et al., 2016)) and that therefore may deviate substantially from this highly idealized situation. It is also possible that complex in-cloud variations lead to multiple local extrema of the radiative heating rate and therefore the  $\mathrm{PVR}_{\mathrm{rad}}$ signal can change sign multiple times within the layer.

Figure 8. (a) Schematic of the vertical temperature tendency profiles due to a radiation-absorbing layer (gray) that mimics a cirrus cloud (inspired by Liou (1986) and Houze (2014)). Black lines show vertical profiles of  $Q_{sw}$  on the left and of  $Q_{lw}$  on the right. The blue and red symbols indicate the sign of the associated radiative PV rate in that layer. Stars denote extrema of  $Q_{lw,sw}$ . (b) Decision tree for column categorization; see text for details.

Table 2. Distribution of different vertical profile types (see Fig. 8) associated with  $PVR_{rad,MAX}$  for TST category IV.

|                        | Top cooling | Base heating | In-layer variation |
|------------------------|-------------|--------------|--------------------|
| Cloudy environment     | 79.6%       | 0.9%         | 1.2%               |
| Cloud-free environment | 17.7%       | 0.3%         | 0.3%               |

In order to analyse the different scenarios leading to positive  $PVR_{rad}$  we separate the different profiles into those where  $PVR_{\rm rad} > 0$  occurs in layer 1 due to longwave cloud-top cooling and those where  $PVR_{\rm rad} > 0$  occurs in layer 4 due to longwave cloud-base heating. Considering both types together would introduce averaging out cloud bases and cloud tops and not result in meaningful composites. To separate the two types, we follow the decision tree shown in Fig. 8b. We consider all time steps with a positive PVR<sub>rad,MAX</sub> in the TST category IV. First, the algorithm aims to identify and remove incloud cases with  $|PVR_{sw}| > |PVR_{lw}|$  in layer 3. After this, we know that the  $PVR_{rad}$  signal can only be produced by a 425 positive vertical gradient of  $Q_{lw}$ . To identify the cloud-base heating cases, we check if there is any longwave heating above the trajectory position. For numerical reasons, we do that by checking for values larger than  $1 \times 10^{-5} \, \mathrm{K} \, \mathrm{h}^{-1}$  in the  $100 \, \mathrm{hPa}$  above the trajectory. If there is no longwave heating, then the  $PVR_{rad}$  signal can only arise from longwave cloud-top cooling. In contrast, if there is longwave heating above, then the signal arises either from cloud-base heating or again in-cloud processes. To separate these, we check if  $Q_{\rm lw}$  changes sign below the trajectory. Once the signal below the trajectory turns negative (which 430 it has to in the case of clear sky below the cloud) it also stays negative. If not, we have a case of in-cloud variation and if yes, the trajectory is experiencing PV production due to longwave cloud-base heating. For the last step, we consider again a layer of 100 hPa, now below the trajectory.

The second step of the categorization determines if the  $PVR_{rad,MAX}$  is from a cloudy or cloud-free environment. We check for higher than  $1 \times 10^{-6} \, \mathrm{kg \, kg^{-1}}$  hydrometeor content in the range of  $\pm 50 \, \mathrm{hPa}$  above and below the  $PVR_{rad,MAX}$  location. The choice of the threshold mimics the limit of what satellites are able to detect, and this has been previously compared to ERA5 in Binder et al. (2020). Combining this two-step categorization provides the values presented in Table 2. A majority of columns get their  $PVR_{rad,MAX}$  from a longwave top-cooling environment (layer 1), as expected. However, 17.7% of the total trajectories are in the top-cooling category with no cloud water or ice content below. In these cases,  $PVR_{rad,MAX}$  is most likely produced on top of a cloud-free moist layer.

# 7.2 How vertical profiles of humidity and clouds shape radiative PVR

We now investigate vertical profiles at times of  $PVR_{rad,MAX}$  for the cases that are linked to cloud-top cooling, separately for cloudy and cloud-free environments. The columns are sorted by the value of  $PVR_{rad,MAX}$  and vertically centred at the pressure level of the trajectory. Then, profiles of different variables are averaged for every bin of  $PVR_{rad,MAX}$  to display the atmospheric conditions below and above the locations with radiative PV rates (Fig. 9). The marginal distributions on top of the main panels reveal that in cloudy conditions the most frequent  $PVR_{rad,MAX}$  values range from 0.02 to 0.08 pvu/h, whereas in cloud-free conditions, the frequency maximum is close to zero and decreases with higher  $PVR_{rad,MAX}$  values. For the cloudy





cases (Fig. 9a-c), the 2 pvu line shows that the high  $PVR_{rad,MAX}$  values occur on average close to the dynamical tropopause. The specific humidity composite for the cloudy environment (Fig. 9a) shows that the larger  $PVR_{rad,MAX}$ , the higher the specific humidity at the level of the trajectory. A similar tendency can be seen for the ice and liquid water content (Fig. 9b, blue and green lines, respectively), where both show higher values with increasing  $PVR_{rad,MAX}$ . In the same panel, the values of  $Q_{rad}$ , including both longwave and shortwave signals, are higher when the trajectory is on top of an optically thicker cloud, and the values are the highest if a mixed-phase cloud is present. The corresponding  $PVR_{rad}$  signal (Fig. 9c) shows the vertical dipole around the vertical temperature tendency gradient, as well as the in-cloud PV destruction as schematically indicated in Fig. 8a. However, we do not observe a cloud-base heating signal in these composites as the clouds can be of very different depths, and the heating signal is most likely averaged out.

Figure 9e shows the cloud species for the TST trajectories in a cloud-free environment. According to our classification (see above), the cloud is 'allowed' to be present  $50\,\mathrm{hPa}$  below the  $\mathrm{PVR_{rad,MAX}}$  level. The  $Q_{\mathrm{rad}}$  signals in this cloud-free environment are smaller in absolute values; however, the corresponding  $\mathrm{PVR_{rad}}$  (Fig. 9f) is comparable, both in amplitude and extent, to that with a cloud. This can be expected because  $\mathrm{PVR_{rad}}$  is determined by the vertical gradient of  $Q_{\mathrm{rad}}$  and by absolute vorticity, and not by the absolute value of the cooling rate. The radiative temperature tendencies in the cloud-free environment are caused by the strong vertical gradient in specific humidity (Fig. 9d) , where the layer with  $25\,\mathrm{to}\,100\,\mathrm{mg\,kg}^{-1}$  is shifted to a lower pressure level (upwards) compared to the cloudy cases. Furthermore, the temperature lines reveal that  $\mathrm{PVR_{rad,MAX}}$  occurs for cloud-free cases in a warmer layer compared to cloudy cases. According to Planck's law, this allows for higher emission and therefore a higher cooling rate.

In summary, the vertical extent of both the cloud-top cooling and the  $PVR_{rad}$  signal ranges from almost  $100 \, hPa$  below to roughly  $25 \, hPa$  above the location of the TST trajectories at the time of  $PVR_{rad,MAX}$ . Based on these column composites, we can conclude that cloud-top cooling, preferably on top of thick clouds, or a strong vertical humidity gradient must be present to enable radiation to substantially modify PV. We emphasize that the most extreme radiative PV rates occur in cloudy situations due to cloud-top cooling above thick mixed-phase clouds. A further analysis of which weather system provides these favourable environments (e.g., warm conveyor belt outflows, e.g., Heitmann et al. (2024)) is beyond the scope of this work and left for future studies.

## 8 Conclusions

This study quantifies systematically and climatologically the role of radiation in PV production leading to TST. We designed a method that combines a Lagrangian identification of TST with another Lagrangian framework to quantify the contribution of different physical processes to the PV modification. Specifically, the focus is on the period along the TST trajectory during which PV increases from less than 1 pvu to the crossing of the dynamical tropopause defined as the 2 pvu surface – we refer to this period as the TST duration,  $\tau_{\rm TST}$ . The PV modifications due to radiation and other processes are traced during this period and the contribution of the accumulated radiative PV tendencies (APV $_{\rm rad}$ ) is quantified for each individual TST trajectory. The established TST identification method resulted in a 10-year (2012-2021) TST climatology based on ERA5 that presents similar

Figure 9. Column composites at  $PVR_{rad,MAX}$  time steps for top-cooling events in the radiatively dominated TST category IV. The large panels show vertical profiles of different fields as a function of  $PVR_{rad,MAX}$  and the small panels on top of every profile composite show the number of cases contributing to each bin of  $PVR_{rad,MAX}$  values. (a, b, c) for cases with a cloud and (d, e, f) for cloud-free cases. The y-axis shows the pressure relative to the vertical position of the TST trajectory at the  $PVR_{rad,MAX}$  time step. From left to right, the panels show in colours (a, d) specific humidity, (b, e)  $Q_{rad}$ , and (c, f)  $PVR_{rad}$ . Overlaid are in (a, d) the 2 pvu contour, and 235 K and 250 K temperature isolines, in (b, e) isolines of frozen hydrometeor water content (sum of cloud ice and snow, in blue) and liquid water content (sum of cloud liquid and rain, in green), and in (c, f) selected temperature tendency contours from (b, e) to guide the eye.







features as previous TST climatologies based on earlier reanalysis datasets. The main conclusions of our study are summarized as follows, sorted according to the research questions posed at the end of the introduction:

- Q1: 84% of the identified TST trajectories experience a positive contribution from radiative PV tendencies (APV $_{\rm rad}$ ). In 25% of the cases, radiation contribute at least a quarter (0.25 pvu) to the PV increase, and in 8% of the cases radiation dominates TST with APV $_{\rm rad}$  > 0.5 pvu. We refer to this TST category as 'radiatively dominated TST' and several detailed investigations in the paper focus on this category.
- Q2: The radiatively dominated TST has a seasonal cycle with highest frequencies in winter and lowest in summer. The spatial climatology of radiation dominated TST shows distinct winter maxima over the central North Atlantic and North Pacific. In comparison to the total TST climatology, the North Atlantic peak of radiatively dominated TST is shifted more inland over Greenland. The spring and summer distributions are more uniform over the Northern Hemisphere, except for a summer peak over the Pan-Tibetan Highlands. The autumn distribution shows a combination of the summer and winter regimes.
- Q3: The way of how  $PVR_{rad}$  is accumulated along the TST trajectories varies greatly in terms of duration and instantaneous values of  $PVR_{rad}$ . The TST duration can range from 10 h to 100 h during which the average  $PVR_{rad}$  and  $PVR_{rad,MAX}$  can vary remarkably from just above 0 pvu to 0.2 pvu (or even higher albeit very rarely). The TST duration does not depend on the radiative contribution itself but mainly on the interplay with other processes. PV modifications due to momentum tendencies (turbulence and convection,  $APV_{mom}$ ) and due to other temperature tendencies than radiation (cloud microphysics, convection, and turbulence,  $APV_{diab-rad}$ ) are faster, i.e., they have higher peak values than those due to radiation. This leaves two options for radiation dominated TST: either there is an extended period with radiative PV production in the absence of other processes (radiation is 'enabled' as an important TST process when there is 'a lack of other processes'), or there is intense radiative PV production when a trajectory enters a region with extraordinarily high values of  $PVR_{rad}$  (see answer to Q4).
- Q4: A systematic analysis of vertical profiles of humidity and cloud species for different intensities of PVR<sub>rad</sub> reveals that for 97% of the trajectories in the radiatively dominated category, maximum PVR<sub>rad</sub> values occur above a longwave cooling signal, induced either by a vertical downward humidity gradient (cloud-free environment, 17.7% of cases) or a cloud-top region (79.6% of the cases). The strongest cooling gradients and highest values of PVR<sub>rad</sub> are produced on top of thick mixed-phase clouds.

There are also a few caveats to keep in mind. The TST identification method does not apply a stratospheric residence time and hence the TST here refers to air that reaches the tropopause and can be subject to stratospheric mixing, but it does not guarantee that the air would stay in the stratosphere. Regarding the process-specific PV rates, we do not quantify the role of momentum tendencies from the model but just expect them to make up the rest of the 1 pvu increase. This means that we do not know the amplitude of the residual or the momentum tendencies separately. The study is also limited to the resolution and the process-modelling capabilities of ERA5. The PV tendencies are only as accurate as the ERA5 temperature tendencies.


This, in turn, depends on how well ERA5 captures the cloud cover in the tropopause region. Given that the radiation schemes in models (including the data in this study) are run at a lower resolution, in both time and space compared to the main core, it could be an important step forward in the future to have more accurate radiative fluxes.

The results of this study highlight the role and importance of the radiative processes in mid-latitude TST. The systematic analysis of radiative PV tendencies has been (to the best of our knowledge) for the first time applied to a large set of TST trajectories diagnosed during an extended time period. We show that radiation can be the leading process for TST and that it also contributes to the required PV production for the majority of TST trajectories, albeit small in amplitude. This confirms that radiation is an essential process for understanding the dynamics of the tropopause region, and especially STE.

*Data availability.* ERA5 reanalyses are available from the Copernicus Climate Data Store (Copernicus Climate Change Service, Climate Data Store, 2023).

Author contributions. TM performed the analyses and wrote the first version of this paper. MS, HW, and HJ helped TM in the design of the study, the interpretation of the results, and the writing of the paper.

*Competing interests.* At least one of the (co-)authors is a member of the editorial board of Weather and Climate Dynamics. The authors have no other competing interests to declare.

*Financial support.* This research has been supported by the Schweizerischer Nationalfonds zur Förderung der Wissenschaftlichen Forschung (grant no. 185049).

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
