# Peer review of "The role of radiation in the Northern Hemisphere troposphere-to-stratosphere transport"

_EGUsphere, 2025_

## Author Comment (AC1)

**The role of radiation in the Northern Hemisphere troposphere-to-stratosphere transport**

Tuule Müürsepp, Michael Sprenger, Heini Wernli and Hanna Joos

January 9, 2026

We are thankful to the reviewers for investing time and effort into helping us further improve the quality of our manuscript. Based on the reviewers' suggestions and comments, the main changes that will be implemented in the revised manuscript are:

- provide a quantitative estimate of the potential temperature change along the TST trajectories.

- re-phrase the conclusions about the role of radiation in TST so that we do not overstate its importance and provide more context.

- elaborate on the outlook and some limitations raised by the reviewers.

This document presents the reviewers' comments in **orange** and our responses in **black.**

**Response to Reviewer 1**

**General comment**

This is a well-written and important paper that examines, from a Lagrangian perspective, the physical processes responsible for potential vorticity (PV) modification during troposphere-to-stratosphere transport (TST), especially during crossings of the 2 PVU dynamical tropopause. Using ERA5-based trajectories together with the diabatic heating rates provided by the reanalysis, the authors quantify the contributions of shortwave and longwave radiation as well as other diabatic processes to PV formation, and compare these with PV tendencies derived directly from ERA5 and interpolated along the trajectories. The residual between the two is interpreted as the contribution from dynamical–dissipative effects. The scientific questions (Q1–Q4) are clearly formulated, thoroughly addressed, and well-summarized in the conclusions. The analysis itself and the high quality of the figures are convincing. The dominant role of clouds and/or strong vertical humidity gradients in producing radiative PV rates is somewhat unexpected but physically plausible and well supported by the results. I have only two major points and a few minor issues listed below.

We appreciate the very positive assessment from Reviewer 1 and thank them for emphasising the importance of our study. We are glad that we have managed to structure the questions and the answers in a way that conveys our method and findings. We will address the detailed and very constructive feedback from Reviewer 1 to further improve the manuscript.

**Major comments**

The total PV change along a trajectory results from both diabatic processes (radiation, latent heating, etc.) and dynamical–dissipative processes (turbulence, wave breaking). In the paper, the radiation-induced PV increase of 1 PVU is linked to a modification of the potential temperature, as expected for a diabatic process. However, the magnitude of the associated cross-isentropic motion is not quantified. Classical conceptual descriptions (e.g. Holton et al., 1995) often emphasize idealized, nearly isentropic pathways for troposphere–stratosphere transport, with PV changes occurring by crossing the tropopause. As I began reading the manuscript, I hoped for a quantitative estimate of the actual $\Delta\theta$ along the radiatively dominated TST trajectories. This seems straightforward to compute from the trajectory data already available. Are the typical $\Delta\theta$ values of order 1 K, 5 K, or 10 K? Including a brief summary of these values would greatly help interpret the physical significance of the radiatively driven PV changes.

We are thankful for this interesting suggestion. We have now calculated $\Delta\theta$ along all TST trajectories as the difference between the last time step and the first time step of the TST duration. The radiatively dominated category shows values ranging roughly from $-7.5\,\mathrm{K}$ to $+2.5\,\mathrm{K}$, with a median of about $-2.5\,\mathrm{K}$ (Fig. 1), revealing that, for this category of TST, radiative cooling dominates over radiative heating. This result nicely agrees with the predominant importance of cloud-top cooling for radiatively dominated TST, as discussed in Sect. 7 (see in particular Table 2). It is interesting to note that when considering potential temperature as the vertical coordinate, these TST trajectories *descend* due to radiative cooling while entering the stratosphere. This non-intuitive behaviour was already noted by Juckes (2000) in his note on "The descent of tropospheric air into the stratosphere". For the other TST categories, where the radiative contribution to the required PV increase is smaller, the median $\Delta\theta$ value gets closer to zero, indicating a symmetric distribution of (relatively weak) radiative cooling and heating, respectively. These TST events can hence be considered as fairly isentropic.

[Figure]

Figure 1: Distribution of $\Delta\theta$ values along the TST duration for different categories of TST as introduced in the manuscript.

You state that 84% of the identified TST trajectories show a positive radiative PV contribution, and that in about 8% of all cases radiation dominates the PV increase prior to TST (i.e. $\Delta PV_{RAD} > 0.5 PVU$). Interpreting these numbers in reverse implies that, for the majority of trajectories, the second term in Eq. (1) — the contribution from dynamical–dissipative processes — is dominant. As much as I appreciate the clarity of your analysis, the manuscript currently gives the impression that radiation is the primary mechanism of PV formation during TST. A clearer statement emphasizing that dynamical–dissipative processes dominate in most cases (with radiation being important but not generally leading) would help balance the interpretation. This clarification would not diminish the relevance of your radiative analysis; rather, it would strengthen the overall message by placing the radiative contribution in the correct dynamical context.

We appreciate the reviewer's comments on this issue. We agree with the point that the manuscript might currently overstate the importance of radiation in TST. We will implement the suggestion and clearly emphasise the role of dynamical-dissipative and other diabatic processes, which provide most of the required PV change along most TST trajectories. This will improve the context of the radiative contribution to TST, which can be the dominant process but is so in less than 10% of all TST events. The wording will be adjusted in the abstract, in Sect. 6, and in the conclusions accordingly.

**Minor comments**

L24
"The amplitude of the net transport is small compared to the upward and downward transports, and in the climatological mean it is upward in the tropics and downward

in the extratropics." I would suggest clarifying that the \*\*net mass exchange exhibits a pronounced seasonality\*\*, and that its amplitude is small compared to the gross upward and downward fluxes. A revised formulation could note that, despite this small net signal, the climatological mean shows upward transport in the tropics and downward transport in the extratropics.

We appreciate this clarifying suggestion and will rephrase the paragraph accordingly.

Eq. (3)
The equation could be written more compactly by using a single summation and grouping the PVR terms in parentheses.

Thank you for noticing such a detail. The equation will be adjusted correspondingly.

Figure 1
This figure is important for understanding the methodology. However, the radiative contribution in this example is very small and confined to a short period (approximately 16–22 h), making it a non–radiation-dominated case. A gentler introduction — also in the caption — would help guide the reader. I recommend explicitly using the accumulated PV notation (APV) in the caption to maintain consistency with the text.

Thank you for noting the importance of this figure to our storyline. We will re-write the caption to be as explicit as possible about the different parts of this figure and will also include the accumulated PV notation from this point on. But actually the figure represents a radiatively dominated case – and it is important for us to realise that this was obviously not clear to the reviewer. The PV and APV are both in units of pvu and are indicated on the left y-axis. For this example, the APV due to radiation is 0.914 pvu at the end of the TST duration. We will add another y-axis on the right side and match the colour code of all the axes to facilitate the interpretation of the figure.

Figure 2 and related text
The definition of the categories used in Fig. 2 should be introduced at the beginning of Sect. 3, so that the reader understands them before encountering the figure. It may also be useful to include a brief reminder of the category definitions directly in the figure caption.

Thank you for pointing this out. Since the results of Fig. 2a 'define' the said categories, it proves a bit tricky to introduce them before the figure. We will change the first mention of Fig. 2 so that it becomes clearer that the definition of the categories arises from this distribution. We will also mention the categories in the figure caption.

Figures 5 and 6
The examples shown in Figs. 5 and 6 should ideally be connected more clearly. For instance, selecting cases with similar TST durations (or another consistent criterion)

would make it easier for the reader to relate the subpanels of Fig. 5 to the corresponding subpanels of Fig. 6.

We apologise, but we don't understand this comment. Figure 5 shows the time evolution of PV and radiative PVR along 4 example trajectories, which span across the phase space shown in Fig. 5a. Figure 6 then shows the 4 identical trajectories in physical space. We believe that the connection between the two figures should be clear.

**Response to Reviewer 2**

**Summary**

This paper presents a climatology of troposphere-to-stratosphere (TST) exchange in the Northern Hemisphere midlatitudes. They find TST from Lagrangian trajectories calculated from ERA5 that increase from 1 to 2 PVU. From these TST trajectories they investigate the physical processes involved in TST, with a particular focus on long-wave radiation. They show that most trajectories (84%) have a positive PV contribution from long-wave radiation and a small fraction have a very strong ($> 0.5$ PVU, 8%) positive PV contribution from long-wave radiation. This contribution can range from a slow accumulation over a long time to more rapid PV production a the tops of sharp humidity or cloud gradients.

This is a well written paper on a interesting subject with novel methods and results, and a good fit for WCD. I would also like to credit the authors on the quality of their figures. They appear to have put a lot of thought into conveying a lot of information in a clear and concise way, and it has worked well.

I have added some minor/technical corrections below. I have also added some discussion that does not need extra work for publishing the paper, but would be good to include in the discussion and consider for future work. In particular, I think that the results are limited by only considering individual trajectories rather than using ensembles of trajectories to look at a full air mass undergoing TST and the total PV changes within it.

We are grateful to Reviewer 2 for their positive comments on the manuscript, and we appreciate the time and effort they have put into the detailed comments and the thoughtful suggestions for further discussion. We also thank them for noticing the work that was put into producing the figures in this manuscript.

**Corrections**

Figure 6 caption mentions red circles, but the circles are black.

Thank you for pointing this out; we will correct the caption.

Thank you for noticing this redundancy. The sentence will be modified as suggested.

The data availability citation for ERA5 says single-level data, but the DOI links to the pressure-level data. However, in the data and methods section you mention the model level data and diabatic tendencies which are available from the MARS archive, not CDS. Can you clarify what data is used to calculate the trajectories and PV tendencies. Is it model level or pressure level data or a mix of both?

Thank you very much for pointing out this discrepancy. All ERA5 data we used was on model levels. The reference should point to the complete ERA5 global atmospheric reanalysis dataset (Hersbach et al., 2017; Copernicus Climate Change Service, 2023); we will fix the link in the revised manuscript.

**Discussion**

It is very noticeable from your case examples that the spikes in PVR$_{rad}$ do not map well to the increases in PV (figures 1 and 5), so there must be some strong cancellations in the PV tendencies along individual trajectories. This has a large effect on what you might consider "radiatively dominated". E.g. in figure 1 more that half of the PV increase from 1 to 2 PVU has happened before there is any significant PV tendency from radiation. I think this is probably because you only consider individual trajectories. I would suggest replacing the word "cases" with "trajectories". E.g. in the abstract you state "Radiation is present in 84% of TST cases". To me, "cases" implies individual events that would be composed of many trajectories. If you were to look at the total contribution of radiation to the complete air mass undergoing TST, you could get a different result for the percentage of events in which radiation has a positive contribution. (My guess is it would be higher). My thought would be that radiation is creating a local sharp PV tendency and turbulent mixing or diffusion is acting to spread it out. If you were to look at a full air mass you might see a strong radiative PV tendency in the middle, cancelled out by negative tendencies from turbulent mixing/diffusion, surrounded by positive tendencies from turbulent mixing/diffusion. Considering the full air mass, you would get a weaker but overall positive contribution from radiation and a roughly zero contribution from the mixing (although it would depend on how it is cut by the 2-PVU limit).

Thank you for discussing and sharing the subtle point about distinguishing trajectories and cases. We agree that since our study uses trajectories only, it would be cleaner to refer to them as 'trajectories' instead of 'cases'. We will take a careful look at the manuscript to make this adjustment consistent throughout the text. Regarding the interplay with the turbulent/mixing processes, we do see some hints of these processes working against

radiation, as shown and discussed in Section 6. The idea of focusing on the evolution of larger airmasses instead of single trajectories might be a sophisticated way of disentangling the role of each of these processes. We agree that the result of this would depend greatly on how the study period is defined (as you mentioned the 2 pvu limit) and whether we could continue to use the TST duration that was defined for the current study. We will definitely keep this point in mind for future work and mention this limitation also in the conclusions.

"The distribution of radiative PV changes along trajectories is skewed because of the selection bias related to our choice of focusing on TST trajectories. If a similar analysis was performed along randomly selected trajectories in the upper troposphere, the radiative PV changes are expected to be symmetrically centred at zero." – The statement about the increased skew due to selection bias makes sense. However, I would still expect upper-tropospheric PV tendencies from long-wave radiation to have some positive bias because you will more often be looking at cloud tops or the tops of humidity gradients.

Thank you for this comment. Indeed, the upper-tropospheric region is favoured to have positive PV tendencies due to the temperature tendencies from long-wave radiation, as long as we consider the cloud tops or humidity gradients. In this sense, the upper troposphere might indeed be positively biased against the PV tendencies due to longwave radiation at lower levels and also against other processes in the upper troposphere. However, what we know from the PVR profile studies (Figures 8 and 9) is that this (rather symmetric) PVR dipole has a narrow vertical extent; therefore, it really depends on the dynamics whether certain trajectories experience positive or negative tendencies and for how long they experience them. It would be interesting in the future to study the climatological PVRs along any trajectories at these levels to place our distribution of the accumulated PVRs along TST trajectories into perspective. We will make sure to mention this in the outlook paragraph.

Your case in figure 5e and 6g,h reminded me of the work of Rodwell et al (2013) (and following papers) on forecasts busts. The location matches up to the region where they highlighted that ascent in MCSs was too weak leading to an underamplified Rossby wave. The timeseries shows a sharp drop in PV which appears to coincide with the 12-hour assimilation cycle. This would be consistent with the assimilation acting to shift the tropopause higher if it was underestimated by the forecast and therefore lowering PV in that region.

Thank you for pointing us to these studies and offering an explanation for the sudden drop in the PV evolution. We are, however, not confident in relating the PV evolution along a single trajectory to general aspects of systematic forecast errors and data assimilation issues.

I was actually fairly surprised how infrequently the data assimilation is leading to spurious TST, from the spikes every 12 hours in figures 4b and 5a.

We had a somewhat opposite reaction at first. A closer look at the Eulerian PVR fields also revealed that the assimilation shifts were larger than expected. This was the main reason why we decided to introduce the two-hour residence time criterion, which evidently reduced the number of these spurious TST events; without this criterion, the spikes in Figures 4b and 5a would indeed be larger.

**References**

Copernicus Climate Change Service: Complete ERA5 global atmospheric reanalysis, Copernicus Climate Change Service (C3S) Climate Data Store (CDS), `https://doi.org/10.24381/cds.143582cf`, 2023.

Hersbach, H., Bell, B., Berrisford, P., Hirahara, S., Horányi, A., Muñoz-Sabater, J., Nicolas, J., Peubey, C., Radu, R., Schepers, D., Simmons, A., Soci, C., Abdalla, S., Abellan, X., Balsamo, G., Bechtold, P., Biavati, G., Bidlot, J., Bonavita, M., Chiara, G. D., Dahlgren, P., Dee, D., Diamantakis, M., Dragani, R., Flemming, J., Forbes, R., Fuentes, M., Geer, A., Haimberger, L., Healy, S., Hogan, R. J., Hólm, E., Janisková, M., Keeley, S., Laloyaux, P., Lopez, P., Lupu, C., Radnoti, G., de Rosnay, P., Rozum, I., Vamborg, F., Villaume, S., and Thépaut, J.-N.: Complete ERA5 from 1940: Fifth generation of ECMWF atmospheric reanalyses of the global climate, Copernicus Climate Change Service (C3S) Data Store (CDS), `https://doi.org/10.24381/cds.143582cf`, 2017.